# Reversing food preference through multisensory exposure

**Avishek Chatterjee**[1,2], **Satyaki Mazumder**[1], **Koel Das**[1] *

**1** Department of Mathematics and Statistics, Indian Institute of Science Education and Research Kolkata, Nadia, West Bengal, India, **2** Department of Neuroscience, University of Minnesota, Minneapolis, Minnesota, United States of America

* koel.das@iiserkol.ac.in

## Abstract

Experiencing food craving is nearly ubiquitous and has several negative pathological impacts prompting an increase in recent craving-related research. Food cue-reactivity tasks are often used to study craving, but most paradigms ignore the individual food preferences of participants, which could confound the findings. We explored the neuropsychological correlates of food craving preference using psychophysical tasks on human participants considering their individual food preferences in a multisensory food exposure set-up. Participants were grouped into Liked Food Exposure (LFE), Disliked Food Exposure (DFE), and Neutral Control (NEC) based on their preference for sweet and savory items. Participants reported their momentary craving for the displayed food stimuli through the desire scale and bidding scale (willingness to pay) pre and post multisensory exposure. Participants were exposed to food items they either liked or disliked. Our results asserted the effect of the multisensory food exposure showing a statistically significant increase in food craving for DFE participants postexposure to disliked food items. Using computational models and statistical methods, we also show that the desire for food does not necessarily translate to a willingness to pay every time, and instantaneous subjective valuation of food craving is an important parameter for subsequent action. Our results further demonstrate the role of parietal N200 and centro-parietal P300 in reversing food preference and possibly point to the decrease of inhibitory control in up-regulating craving for disliked food.

## Introduction

Food preference and consumption constitute a vital aspect of our daily life and have a long-term impact on our health. Food craving, signifying a strong liking for a specific food item, has recently garnered a lot of attention due to its effect in the field of psychology, behavioral economics, and medicine. Craved items are highly palatable and energy-dense comprising of high fat and/or sugar content [1]. Pathologically, food craving contributes to various negative outcomes including obesity, addiction and other disorders like binge eating disorder (BED) and bulimia nervosa [2]. Obesity, in particular, deserves special attention due to its unprecedented increase globally. Most of the public health interventions to sensitize people to healthy food choices in order to control obesity seems to be ineffective. Ultimately choosing to eat healthy

**Data Availability Statement:** A minimal dataset is available for download at https://zenodo.org/record/8019346. Entire data cannot be shared publicly because of privacy concern and specific lack of consent for making data public. However data are available from the corresponding author

(koel.das@iiserkol.ac.in) for researchers who meet the criteria for access to confidential data.

**Funding:** A. Chatterjee was supported by an INSPIRE fellowship (no: IF170367) from the Department of Science and Technology (DST), Government of India. The funders had no role in study design, data collection and analysis, decision to publish, or preparation of the manuscript.

**Competing interests:** The authors have declared that no competing interests exist.

food requires self-control over our thought and behavior and effective cognitive strategies can be devised to facilitate self-control. Previous research has demonstrated that cognitive strategies reduce craving [3–6] and reduction in craving for non-nutritious food has been shown to be a precondition for healthy eating behavior [7, 8].

Developing efficient strategies to control food craving has gained considerable attention from public health perspectives due to an increase in craving-related disorders in the last decade [6, 9–12], but unfortunately practical applications of efficient intervention strategies remain limited. Up-regulation and down-regulation of craving tendencies typically use cognitive regulation strategies [9, 13, 14]. In behavioral economics, "nudging" approach has been shown to guide people towards healthy food choices by strategies including controlling portions and having healthy default food options [15–18]. However, the applicability and generalizations of earlier offline nudging approaches in real-life remains limited [19]. Recent studies using digital nudging [20–22] seem effective and underscore the need to explore the neural mechanism guiding up-regulating of nudged food items. Strategies for up-regulating or reversing food choice can be potentially used to develop effective public health policies to control eating-related disorders [7, 8].

The effect of craving is typically transient and progress from craving to valuation is best captured momentarily [23]. Although craving is known to modulate our valuation system, the effect of craving on consumer behavior remains unexplored in most studies where typically craving is measured by self-reported inconsequential desire rating for the displayed food cues [24–26].It is important to notice whether increased liking for a particular food is strong enough to elicit an objective valuation of that food which ultimately translates to the consumer spending financial resources in procuring the desired food.

Event-related potentials (ERPs), with their high temporal resolution, allow identifying the time course of cortical processes of related events with a specific stimulus. Assessing initial sensory or visual attention in response to food cues can elucidate further higher-order attentional processing, which may impact subsequent eating behavior, [27–31]. Earlier work on neural modulators of food choices demonstrate the role of P300 and LPP [32] for food cues in both obese and normal weight individuals. One of the early works [29] exploring the attentional processing of food-related words between normal and obese individuals found enhanced P200 for obese participants and food bias for P300 in all participants showing enhanced early attentional allocation for food in obese individuals. Early attentional bias to food items especially for high calorie food has been displayed by obese individuals [28, 29, 33, 34]. The ERP components P200/P2, and posterior N200/N2 onset are associated with initial sensory or visual attention (for review, see [35]). Enhanced oriented attention to food stimuli has been displayed by obese individuals using P200 [29] Greater P2 (or P2 like) amplitude enhancements are associated with food stimuli as compared to neutral stimuli [30] and with chocolate stimuli as compared to non-chocolate stimuli in the group of female chocolate cravers [36]. P300 and late positive potential (LPP) are two commonly examined components that reflect higher-order attention and the motivational relevance of a stimulus and its emotional valence and are typically modulated by palatable and meaningful food cues [35]. P300 is commonly associated with automatic attention allocation [37] and inhibitory control [38]. Several studies reported higher P300 amplitude to food than non-food stimuli [32, 39] signifying greater attention allocation of food than non-food stimuli. In the addiction literature, less pronounced P300 amplitude in addicted populations in comparison to controls is considered as a marker for neural deficits in inhibitory control [38]. LPP on the other hand reflects more extended attention allocation due to the emotional and motivational relevance of pictures which lead to higher amplitudes corresponding to the more arousing stimuli [35, 40].

The aim of the proposed study is to systematically study the effect of food craving through psychophysical tasks. Specifically, we explore the following aspects:

- Translation of behavioral desire ratings for craved foods to Willingness-To-Pay value measures.

- Prediction of Willingness-To-Pay value for craved food items

- Feasibility of the reversing food preference by inducing craving for disliked foods.

- Neural correlates of food craving and its modulation (univariate and multivariate analyses).

In the current study, following the experimental procedure of a previously published study [23], using both a desire scale and a bidding scale, we measured the instantaneous desire as well as the willingness to pay (WTP) for the food items. WTP is a common measure in the field of behavioral economics to study consumer behavior and helps translate the desire for food into subjective valuation [6]. We wanted to explore whether the desire for a craved food item always translates to WTP and whether it is possible to predict momentary valuation for a given food stimulus. We used a detailed statistical model to predict WTP and to understand the role of related external factors on the subjective valuation of craved food.

One of the primary objectives of the current study is to induce a reversal of craving preferences by exposing human observers to food items they dislike using multi-sensory food exposure. Most study design uses the same food items as cues/exposed food to participants irrespective of their preferences [23, 24]. In many studies [23, 41], popular high calorie food items are displayed/exposed to the participants randomly to induce craving. Ideally, the inducement of craving and its effect would depend on the participant's desire for that particular food item. Imagine a scenario where the participant is exposed to a savory item for the inducement of craving when she/he dislikes savory items and prefers sweet items. It would probably not generate the desired effect and will produce erroneous results and confounding interpretations. In this study, participants' food preferences were taken into account while exposing them to the food items to explore their craving tendency. Participants were grouped into three categories based on their preferred food type and exposed food was manipulated to explore the effect of craving for liked and disliked food following multisensory exposure within each group. Unlike many studies that down-regulate craving [5, 42], we wanted to study whether it is possible to up-regulate one's liking for previously disliked food items by use of multisensory food exposure.

Finally we explored neuropsychological correlates elucidating the underlying mechanisms guiding craving using univariate and multivariate analysis. In the current study, we compute the known ERP components typically used in craving literature and explore the early and late neural markers that modulate craving for liked and disliked food items.

## Materials and methods

### Ethical statement

Written informed consent was obtained from each participant before the experiment. The experimental protocol was in accordance with the Declaration of Helsinki and approved by the the Institute Ethics Committee of IISER, Kolkata (No: IISERK/IEC/2021/04).

### Participants

Ninety-six adults (*n* = 96, 49 female and 47 male, ages: 18–29, mean: 21.56, std: 2.64) participated in two experiments, 57 participated in the Behavioral Experiment and 39 in the EEG

Experiment. All participants were healthy with normal weight (18.1 < *BMI* < 29.9), non-dieters with normal or corrected-to-normal vision, and gave informed consent prior to their participation. Exclusion criteria for all participants were: being dieters (medically or self-imposed dietary restrictions), having any history of eating disorders, having any history of chronic illness, being younger than 18 years, and being obese (Body Mass Index [BMI] $\geq$ 30) or underweight (BMI < 18).

Time duration refrained from eating ($\geq$ 2 hours), before coming to the lab was reported by all the participants. The median of such time duration was calculated and based on the median value, two classes, shorter duration, and longer duration were formed. Participants depending on their time from their last meal were sorted into two classes (see details in the S1 File). A participant is excluded from the experiment if his/her time duration of refraining from eating before the experiment is either less than two or more than six hours. The estimated sample size using power = 0.8 and estimated effect size ($\eta^2$) = 0.35 comes to 18 using the more power software [43] and we have an average of 19 participants per group for our task.

Indian Institute of Science Education And Research Kolkata's Ethics Committee approved all procedures (Ethics Protocol No: IISERK/IEC/2021/04).

## Neuroeconomic decision-making task

The pre-questionnaire of participant selection contains likert scales [44] for ten food items (five chocolates and five chips items). Each likert scale ranges from the value 1 to 10, where 1 and 10 indicate the minimum and maximum desire for the item, respectively. Based on the pre-questionnaire response (collected several days prior to the experiment), participants were exposed to either one of the food items in the multisensory food exposure of the experiment and thus divided into three groups, namely Liked Food Exposure, LFE (exposed to a particular food item with specific liking), Neutral Control, NEC (exposed to chocolate or chips, since they are neutral to both food items) and Disliked Food Exposure, DFE (exposed to a particular food item with specific disliking). The details of the above class divisions are mentioned in the Supporting Information.

The participants were presented with visual stimuli consisting of liked and disliked food items, followed by multisensory exposure to a food item. Participants were again shown visual stimuli following food exposure. On finishing the experiment, the participants answered a post-questionnaire. The experiment lasted approximately one hour, which was realized in one session. The experiment was always conducted on the afternoon between 3–5 p.m. Participants had to refrain from eating or drinking anything (except water) for at least two hours prior to the experiment to increase motivation for the sweet and savory food items offered. We noted the time of their last meal at the beginning of the experiment. All participants entered their responses in the Food Craving Questionnaire-State [45, 46] (FCQ-S; Fig 1A) before starting and after completing the experiment. FCQ-S is a 15-item well founded and definitive measure of state fluctuations in self-reported food craving. Responses for each of the questions were scored on a likert scale from 1 to 5 and a total state Food Craving score, which ranges from 15 to 75 was calculated. High scores indicate a strong food craving experience and low scores indicate a low food craving experience.

**Behavioral experiment.** The behavioral data was collected for 17 DFE, 19 LFE and 21 NEC participants making a total number of 57 participants. The task included a total of 15 blocks (2 practice + 3 preexposure + 10 postexposure blocks; Fig 1A). The first two practice blocks were not considered in the analysis of the experiment. Each block consisted of 10 trials (5 chocolate + 5 chips), leading to a total of 130 trials (excluding the practice trials). Each trial consisted of one high-resolution color image of a food item shown for 1 second followed by a

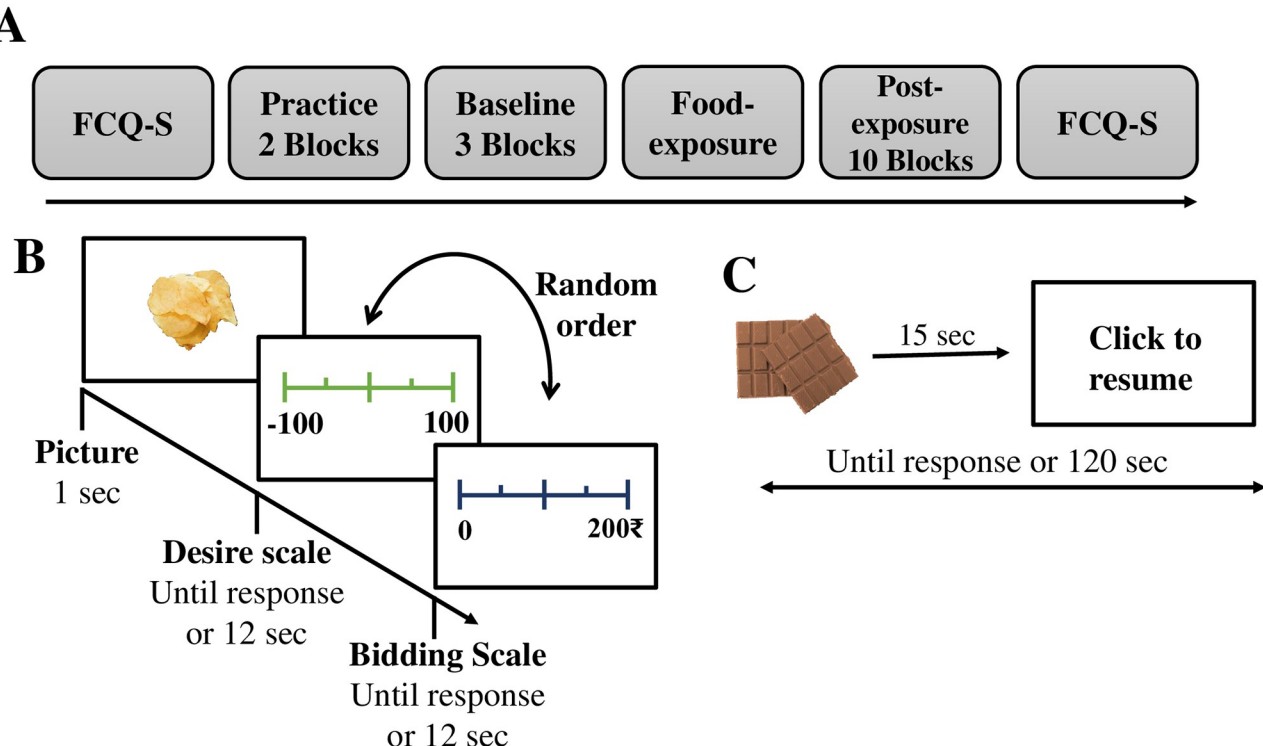

**Fig 1. Experimental task.** A: Timeline of the Experimental procedure. B: Example of one trial consisting food item, desire scale and bidding scale. C: Maximum two minutes of multisensory exposure. Participants are asked to sense the food item but not consume it.

desire scale shown in green color for 12 seconds or until response and a bidding scale in deep blue (Fig 1B) with similar duration as desire scale. Both the scales were mouse-controlled with a slider provided to help reach the required response. The desire scale ranged from the value -100 to 100, where -100 and 100 stand for the minimum and maximum desire for the food item respectively. The order of appearance of the desire and the bidding scale were set randomly (Fig 1B) Participants were given a virtual ₹200 endowment for each block, with which they could bid for the food items. Thus, the range of the bidding scale was dynamic and changed with each trial. It ranged from 0 to the amount left for the rest of the trials (i.e., ₹200 -the amount spent till the trial) in a particular block. Desire and bidding scales were assigned in a random order in a trial so that participants could not predict the appearance of either of the scales. Participants were explicitly instructed to rate their subjective value in the form of a virtual monetary bid value and their concurrent desire for the displayed food item at that moment.

Each participant experienced multisensory food exposure for a maximum of two minutes (Fig 1C) after completion of three preexposure blocks and before resuming the remaining 10 postexposure blocks (Fig 1A) to induce food craving. Participants were given a real food item (chocolate or chips) according to their class levels (i.e., DFE, LFE, or NEC) and instructed to sense the food item by smelling and seeing it but without having it. The whole experiment was monitored through a camera from outside the experiment room especially to confirm that participants refrained from eating at the time of multisensory exposure. Participants received ₹100 and a snack food item that he/she desired the most for their participation and a bonus amount, which was calculated using the standard Becker-DeGroot-Marschak (BDM)

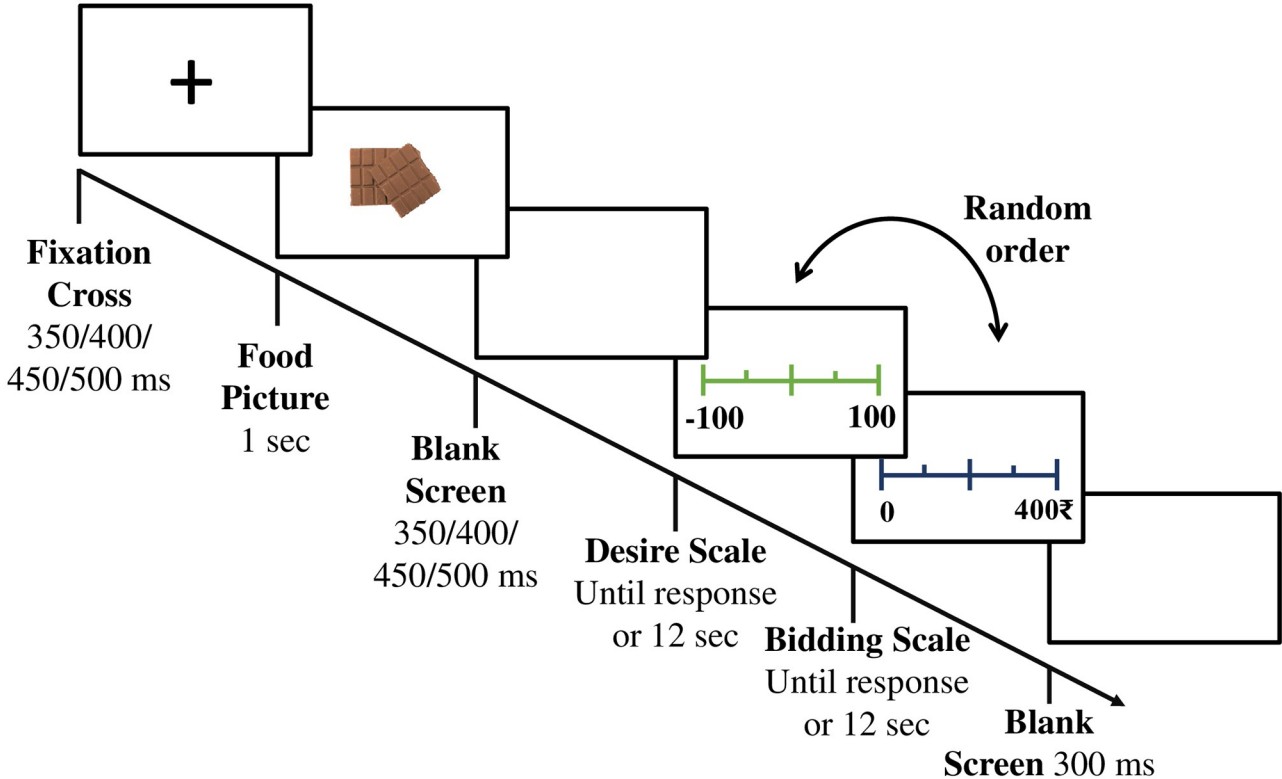

**Fig 2. Example trial of EEG data collection.** A fixation screen with center cross was displayed with a jittered time window (350–500 ms) before the appearance of food stimuli. Two additional blank screens were added. First one was after the stimuli was shown and the other one was at the end, i.e., after the 2nd response screen.

mechanism [47]. Instead of considering BDM on a single trial from the last completed block [23], we randomly selected one out of 13 blocks and considered all trials in that block for the bonus calculation using standard BDM auction (see Supporting Information for details).

**EEG experiment.**   A second experiment was conducted, which included EEG recordings while the participants performed a similar neuroeconomic task. The experimental paradigm for the EEG experiment is given in Fig 2. In the second experiment, EEG data were collected for 13 DFE, 13 LFE, and 13 NEC participants making a total number of 39 participants. Data from four participants (1 DFE + 1 LFE +2 NEC) was excluded from the analysis due to faulty recording. In the EEG experiments, 10 additional trials were used in each of the blocks, which resulted in 10 chocolate and 10 chips trials per block. Participants were given a virtual endowment of ₹400 for each of the blocks for the bidding. Participants' compensation details are mentioned in the supporting information. Apart from the additional trials and slight modification in the experimental paradigm to account for better neural data, all other aspects of the EEG experiment remained the same as the behavioral experiment.

## Neural data acquisition and preprocessing

EEG activity was recorded using 64-channel active shielded electrodes mounted in an EEG cap, which follows the international 10/20 system. We have used two linked Nexus- 32 bioamplifiers at a sampling rate of 512 Hz to record the EEG signals. Trials were epoched using the available trigger information and epoched trials were preprocessed using the EEGLAB toolbox [48]. Trials were band-pass filtered (0.01—50 Hz.), baseline corrected, and referenced

using average referencing. Trials with ocular artifacts like blinks and eye movements were detected using bipolar electro-occulograms (EOG) and visual inspection and trials with amplitude exceeding ± 100 mV were excluded from the analysis. 3.6% of the trials were discarded as outliers and not used in the analysis.

## Behavioral data analyses techniques

**Preliminary analyses.** We performed a couple of analyses prior to and post to the main experiment to deduce the state of craving and time dependency between participants' last meal taken. Inducement of cravings in participants was measured using the FCQ-S values before and after the completion of the experiment. In order to compare the FCQ-S value before start and after completion of the neuroeconomic decision-making task, the total FCQ-S value was computed for each of the participants by adding the response value (between 1–5) for all 15 questions. Further, paired t-test was performed between before and after conditions for each of the participant categories to check whether there was a significant increase in FCQ-S value at the end of the experiment.

Time dependency between participants' last meal taken and inducement of craving was checked using frequency chi-square ($\chi^2$) tests of independence based on a $2 \times 2$ contingency table. To perform this analysis we formed two classes, namely, shorter duration and longer duration, as mentioned in the Participants section.

We then calculated the difference ($d_i$) between FCQ-S values (after-before) for each participant. Participant $i$ was classified in YES Craving class if $d_i > \frac{s}{\sqrt{n}} t_{(n-1),\alpha}$ else in NO craving class, where $n = 57$ (Number of participants), $s$ = standard deviation and $t_{(n-1),\alpha}$ is the $(1 - \alpha)$th quantile of t-distribution with $(n - 1)$ degrees of freedom. In our analysis, we have chosen $\alpha = 0.05$.

**Change point detection.** We initially used the change point detection method as an exploratory analysis to find any significant change in the time series data following multisensory exposure. Change point analysis [49] is used to identify and estimate times when the probability distribution of a stochastic process or time series changes. In the experiment, we tried to identify whether any change point exists in the exposed time series (time series of desire and bid-values for exposed food items). We checked whether the means of these time series (taken over participants' responses at each time point) had any sudden change at any point of time. We used the AMOC method [50] to detect change points for both bid value and desire value time series for the exposed food item.

**Computational model.** In order to predict the bid value (representing WTP) which provides the subjective valuation of exposed food item per trial, we used a computational model which enables us to analyze the effect of several factors on WTP in a detailed manner. Computational modeling was performed to predict the bid value based on different explanatory variables as desire values, reaction time, exposed food type, and their interactions. We have considered the following three models:

- Model 1: Considering desire values and reaction time as covariates.

- Model 2: Considering desire values, reaction time, and exposed food type as covariates.

- Model 3: Considering desire values, reaction time, exposed food type, and its interaction with desire values as covariates.

Taking into account the AIC and BIC values, we have included the optimal model in the main manuscript and kept the results of the remaining models in the Supporting information. While formulating models for DFE, LFE, and NEC participants incorporating spline

regression was needed as the respective scatter plots of bid value and desire rating indicate (see S1 Fig in S1 File). The main idea of a spline regression model (SRM) is to divide the range of the *x* (explanatory variable) into segments and fit appropriate spline functions in each of the segments. Splines are piece-wise polynomials of order *k* and the joint points of the segments are generally termed knots (see S1 File for more details). In our case, the knots were determined on the range of desire value for each of the three models (mentioned above). Knots of SRM were determined following the results of scatter plot (S1 Fig in S1 File). In order to reject the outliers from the data, we have used Cook's distance method [51] for each of the models. Details of our model can be found in the Results section.

## Neural data analyses techniques

**Univariate analysis.** In order to find out the differential ERP amplitude between preexposure and postexposure conditions for the exposed food item, we investigated P200, N200, P300, and Late Positive Potential (LPP) for all participant categories. P200 was measured at the frontal cluster (F1, F2, F3, F4, Fz, AF3, AF4) around 200 ms after stimulus onset. N200 was measured at the parietal electrode cluster (P1, P2, P3, P4, Pz, PO3, PO4) in an early time window around 200 ms after stimulus onset. P300 and LPP were measured at the centro-parietal electrode cluster (CP1, CP2, CP3, CP4, CPz, P1, P2) around 300 ms and 300 to 600 ms time window respectively. Peak amplitudes were identified for all the ERP components except LPP. In LPP, the mean amplitude in the 300 to 600 ms time window was measured. To compare the ERP components between preexposure and postexposure conditions, paired sample t-tests were computed for each of the participant categories.

Partial correlations were calculated for each group at both preexposure and postexposure conditions to find out the relation between dependent measures, i.e., between subjective desire ratings and ERP components (P200, N200, P300, and LPP). Alike other studies [24, 52, 53], $r > 0.24$ (corresponds to $d > 0.5$) and $r > 0.3$ (corresponds to $d > 0.8$) were considered as moderate and large correlation, respectively. Further, multiple testing followed by Benjamini and Hochberg [54] p-value correction was employed.

**Multivariate analysis.** Univariate EEG analysis is popular and widely used to establish the relationship between behavioral performance and neural activity in several cognitive tasks. However, the univariate EEG analysis techniques fail to completely employ the spatio-temporal structure of the multivariate neural data. Applications of multivariate pattern analysis (MVPA) techniques help to incorporate the spatial and temporal information present in the EEG data by integrating the neural information into a single decision variable. Successful application of MVPA has been depicted in numerous studies using EEG and fMRI ([55–57]). Since EEG data is high dimensional and often suffers from the small sample size problem we have used Classwise Principal Component Analysis (CPCA) [58] to classify preexposure and postexposure classes for all participant categories. CPCA, a supervised MVPA technique has been successfully used in previous studies [57, 59–63] to reduce the dimensionality of the EEG signals and extract informative features. The technique is based on the application of principal component analysis (PCA) in each of the classes and aims to identify and discard the non-informative subspace present in the data. The classification is then carried out in the residual space in which small sample size problems and the curse of dimensionality no longer hold. Classification for single-trial EEG data was computed using Linear Bayesian Classifier. Pattern analysis was performed using the leave one out trial cross validation method for each of the individual participants and mean classification accuracies for all participant categories are reported in the results. Further, a t-test was performed to check whether the classification accuracy is significantly above chance.

We have primarily used parametric testing procedures for both behavioral and neural data analysis since more than 80% of the data sets satisfy the normality assumption (using Shapiro Wilk test [64]). Additionally, we have also provided Bayes factors to justify the statistical findings wherever applicable.

## Results

### Behavioral data analysis

**Time duration refrained from eating.** The chi-square test to check the effect of time duration since last meal on craving reveals (Table 1) that there is no reason to believe that craving is dependent on the time since last meal ($\chi^2(1) = 0.6121$, $P = 0.43$). Similarly, the chi-square test for the EEG experiment shows the same result ($\chi^2(1) = 0.2292$, $P = 0.6321$, see S1 Table in S1 File and the subsection followed by the table in S1 File for details).

**FCQ-S value.** We have performed paired-sample $t$-test for all participant categories, to compare measured FCQ-S values before the start of and after completion of the experiment. In all the participant categories (DFE, LFE, and NEC), FCQ-S value increased after completion of the experiment (Fig 3A), indicating an inducement in a food craving-like state throughout the experiment. The increment in mean FCQ-S value was found to be significant for all groups (DFE: $t_{16} = 2.71$, $p-$value $= 0.0077$, $d = 0.66$, Bayes Factor($BF$) $= 7.4338$; LFE $t_{18} = 4.34$, $p-$value $= 1.9690 \times 10^{-4}$, $d = 1$, $BF = 166.78$; and NEC: $t_{20} = 4.56$, $p-$value $= 9.4735 \times 10^{-5}$, $d = 1$, $BF = 312.85$). Similarly, a significant increment in FCQ-S value after completion of the experiment was observed for all participant categories performing the EEG experiment (see Fig 3B); DFE: $t_{11} = 3.18$, $p-$value $= 0.0044$, $d = 0.92$, $BF = 13.07$; LFE: $t_{11} = 4.03$, $p-$value $= 9.91 \times 10^{-4}$, $d = 1.16$, $BF = 44.91$ and NEC: $t_{10} = 2.52$, $p-$value $= 0.0151$, $d = 0.76$, $BF = 4.9172$).

**Change point detection.** Bid-value series of the DFE group showed one single change point at the sixteenth trial (the first trial of postexposure) with the estimated bid value mean 1.0247 and 2.4352, respectively, before and after the change point (Fig 4A). But no change point was detected, with the estimated bid value mean 1.1851 and 1.3153 for both LFE and NEC bid value series, respectively, (Fig 4B and 4C). Similarly, the desire values of the DFE group showed one single change point at the fifteenth trial with the estimated mean desire value −0.2941 and 0.9102, respectively, before and after the change point (Fig 4D), and no change point was detected for LFE desire series with 1.0199 as the estimated mean (Fig 4E). But unlike the bid value series, desire value series for the NEC group showed one single change point at the fourteenth trial, with the estimated mean 0.6946 and 2.0269, respectively, before and after the occurrence of the change point (Fig 4F). Further, change point detection analysis on desire values of non-exposed food items showed a single change point at the twelfth trial only for DFE with estimated desire values 1.1669 and 0.0667, respectively, before and after the occurrence of the change point (see Fig 5), thus, showing a reduction in desire for previously liked food.

Therefore, using the change point detection technique, we observe that DFE participants showed one single change point after multisensory food exposure for both exposed desire

**Table 1. Time duration difference in response to question of whether food cravings had ever been experienced.**

|  | Shorter duration | Longer duration | Test Statistic |
|---|---|---|---|
|  | $n = 29$ | $n = 28$ |  |
| YES craving | 66% (19) | 75% (21) | $\chi^2(1) = 0.6121$, $P = 0.43$ |
| NO craving | 34% (10) | 25% (7) |  |

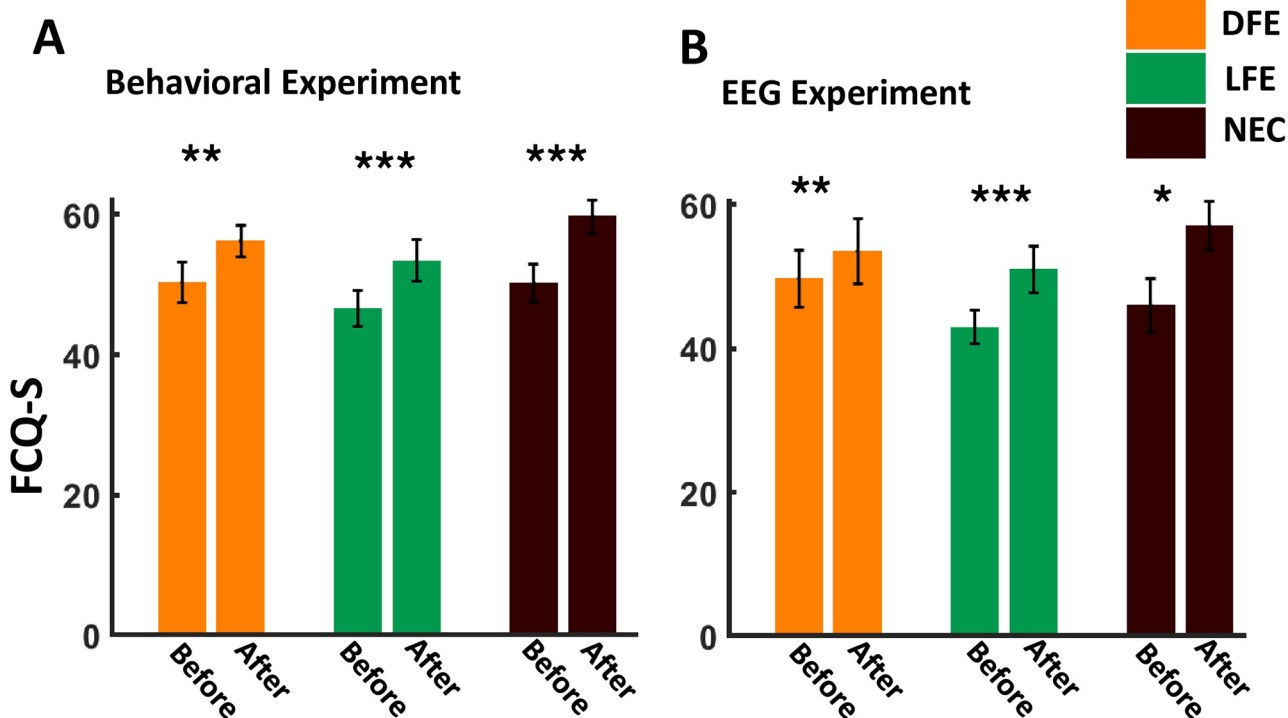

**Fig 3. FCQ-S values (means ± SEM) at before start of and after completion of the experiment.** A. FCQ-S values for the behavioral experiment. B. FCQ-S values for EEG experiment. The figure depicts an increment in FCQ-S value after completion of the experiment for all participant categories in both the experiments. $^{*}p < 0.05$, $^{**}p < 0.01$, $^{***}p < 0.001$.

value and exposed bid values. Also, NEC participants showed one single change point only for the exposed desire value after the multi-sensory exposure but not for bidding value (WTP). Moreover, the estimated mean values increased at postexposure than preexposure for all such observed change points. However, LFE participants did not show any change points for both exposed desire and bid value series. Further, DFE participants showed one single change point after multisensory food exposure for the non-exposed desire value (Fig 5). Moreover, the estimated mean desire value for the non-exposed food at postexposure decreased than the preexposure mean value, thus, showing a reduction in desire for previously liked food.

### Computational model

We have used three types of models to predict bid values (see Supporting information), and based on the AIC and BIC values the optimal model (Model 1) is selected. AIC, BIC values, Adjusted R-squared values, and significant covariates of each of the three models are provided in the Supporting information (see S2 Table in S1 File).

**For DFE.** Change point detection analysis confirmed a significant change in the mean value of preexposure compared to the postexposure blocks for response corresponding to exposed food items. So, we have considered two distinct spline regression models (SRMs) for preexposure and postexposure blocks. The preexposure (1) and postexposure (2) models of

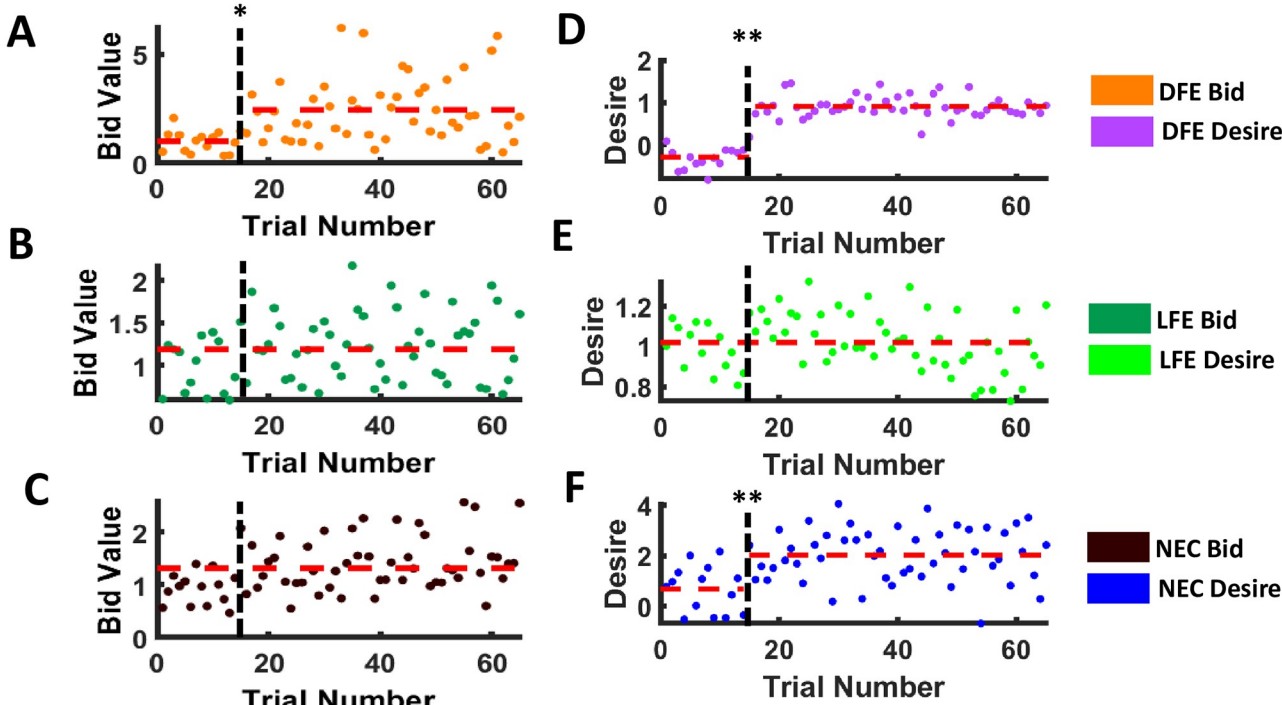

**Fig 4. Change point detection for exposed series.** Scatter plot of all participant categories for both bid value and desire value exposed foods. Vertical black dashed lines indicate trial 15, after which multisensory food exposure has occurred. Red dashed lines indicate estimated mean value and a jump in that line indicates the occurrence of a change point. This clearly depicts that both bid value and desire value exposed series showed a single change point at sixteenth and fifteenth trail respectively for DFE, no change points for LFE and only one change point at fourteenth trial for NEC desire value exposed series. $*p < 0.05$, $**p < 0.01$, $***p < 0.001$.

DFE participants can be written as follows:

$$BV_{\text{Pre}} = (\alpha_0 + \alpha_1 DV_{\text{Pre}})I(DV_{\text{Pre}} < 0) + (\alpha_2 + \alpha_3 DV_{\text{Pre}})I(DV_{\text{Pre}} \geq 0) + \alpha_4 RT_{\text{Pre}} + \epsilon \qquad (1)$$

$$BV_{\text{Post}} = (\beta_0 + \beta_1 DV_{\text{Post}})I(DV_{\text{Post}} < 0) + (\beta_2 + \beta_3 DV_{\text{Post}})I(DV_{\text{Post}} \geq 0) + \beta_4 RT_{\text{Post}} + \epsilon, \quad (2)$$

where $BV_i$, $DV_i$ & $RT_i$ denote the bid value, desire value, and reaction time of bid value, respectively, at preexposure or at postexposure depending on $i$ = Pre or Post, and $\alpha$'s and $\beta$'s are the coefficients of the model (1) and (2) respectively. Preexposure model turned out to be significant ($F_{5,232} = 142.1$, $P < 2.2 \times 10^{-14}$, Adjusted R—squared = 0.75) and all but $\alpha_1$ (coefficient of desire value in [-100,0)) and $\alpha_4$ (coefficient of reaction time) were significant (see Table 2). To check the predictive performance of our proposed model, we estimated the predictive band and depicted in Fig 6A. As clearly seen, the majority of the true values lie within the 95% predictive bands.

Also, the postexposure model appeared to be significant ($F_{5,560} = 560.1$, $P < 2.2 \times 10^{-16}$, Adjusted R—squared = 0.83) and all the coefficients except $\beta_4$ (coefficient of reaction time) had a significant effect in the model (see Table 3). The predictive performance of our proposed model is checked using the predictive bands and is given in Fig 6B and 6C. Most of the true values lie within the 95% predictive bands. Hence, no effect of reaction time on bid value was observed for both preexposure and postexposure models. In the DFE preexposure model, the non-significant $\alpha_1$ showed no dependency between bid value and desire value for the non

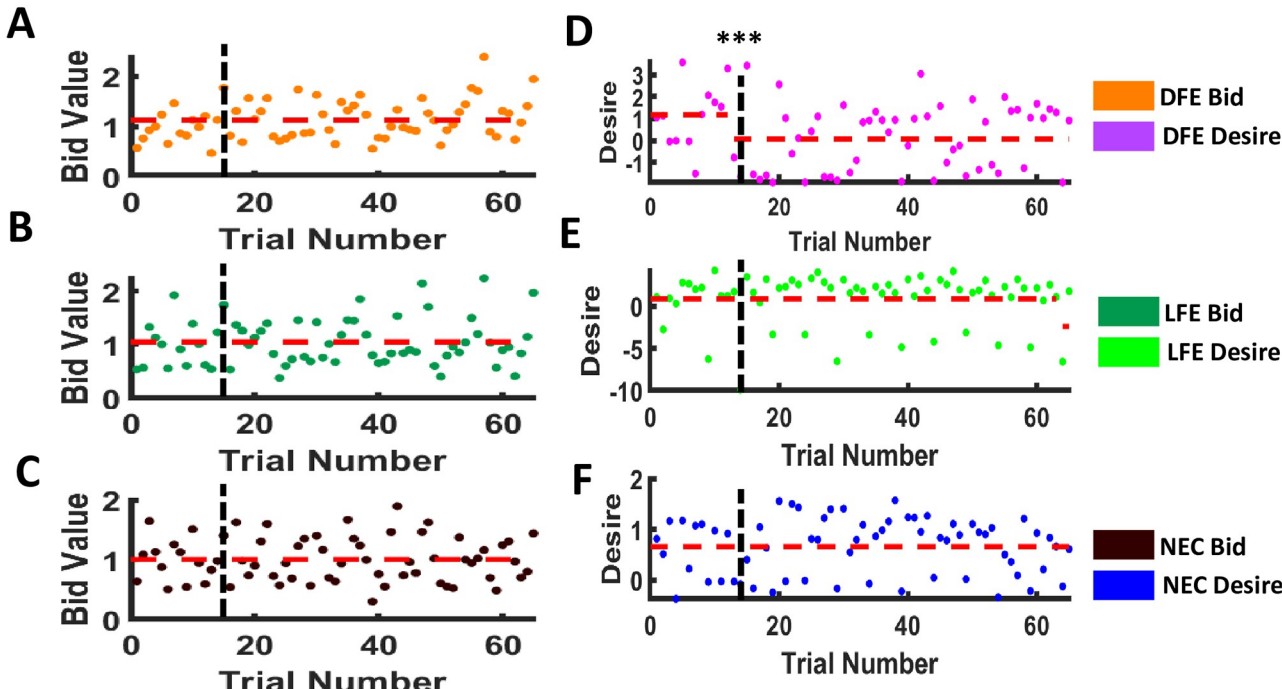

**Fig 5. Change point detection for non exposed series.** Scatter plot of all participant categories for both bid value and desire value exposed foods. Vertical black dashed lines indicate trial 15, after which multisensory food exposure has occurred. Red dashed lines indicate estimated mean value and a jump in that line indicates the occurrence of a change point. This clearly depicts that the desire value non exposed series showed a single change point for DFE, and no change points for LFE and NEC were detected after multisensory food exposure. ***$p < 0.001$.

desired food items. But $\beta_1$ was significant in the DFE postexposure model, which implies an effect of desire value on bid value for the non desired food items postexposure.

**For LFE.** Change point detection analysis confirmed no significant change in the mean value at preexposure compared to the postexposure blocks for both bid value and desire value of the exposed series, which drove us to consider only one spline regression model for the entire exposed series. Alike the DFE models, the knot was taken at desire zero. Our proposed LFE model (3) can be written as follows:

$$BV = (\gamma_0 + \gamma_1 DV)I(DV < 0) + (\gamma_2 + \gamma_3 DV)I(DV \geq 0) + \gamma_4 RT + \epsilon, \tag{3}$$

where $BV$, $DV$ & $RT$ denote the bid value, desire value, and reaction time of bid value, respectively, for the LFE group and $\gamma$'s are the coefficients of the model. This model turned out to be

**Table 2. Spline regression model results.** $\alpha_0$ and $\alpha_2$ are the coefficients of the intercepts when desire value is in [-100,0) and [0, 10] respectively. $\alpha_1$ and $\alpha_3$ are the coefficients of the desire value in [-100,0) and [0, 10] respectively. $\alpha_4$ is the coefficient of reaction time. Coefficients with highlighted p-values are significant in the model. *$p < 0.05$, **$p < 0.01$, ***$p < 0.001$.

| Spline regression model (DFE preexposure). | | | |
|---|---|---|---|
| **Coefficients** | **Estimate** | **t-value** | **p-value** |
| $\alpha_0$ | 40.54 | 2.638 | 0.0089 ** |
| $\alpha_1$ | −0.20 | − 1.193 | 0.2342 |
| $\alpha_2$ | 87.619 | 8.066 | $3.88 \times 10^{-14}$ *** |
| $\alpha_3$ | 1.7617 | 7.646 | $5.47 \times 10^{-13}$ *** |
| $\alpha_4$ | 0.0002 | 0.098 | 0.9218 |

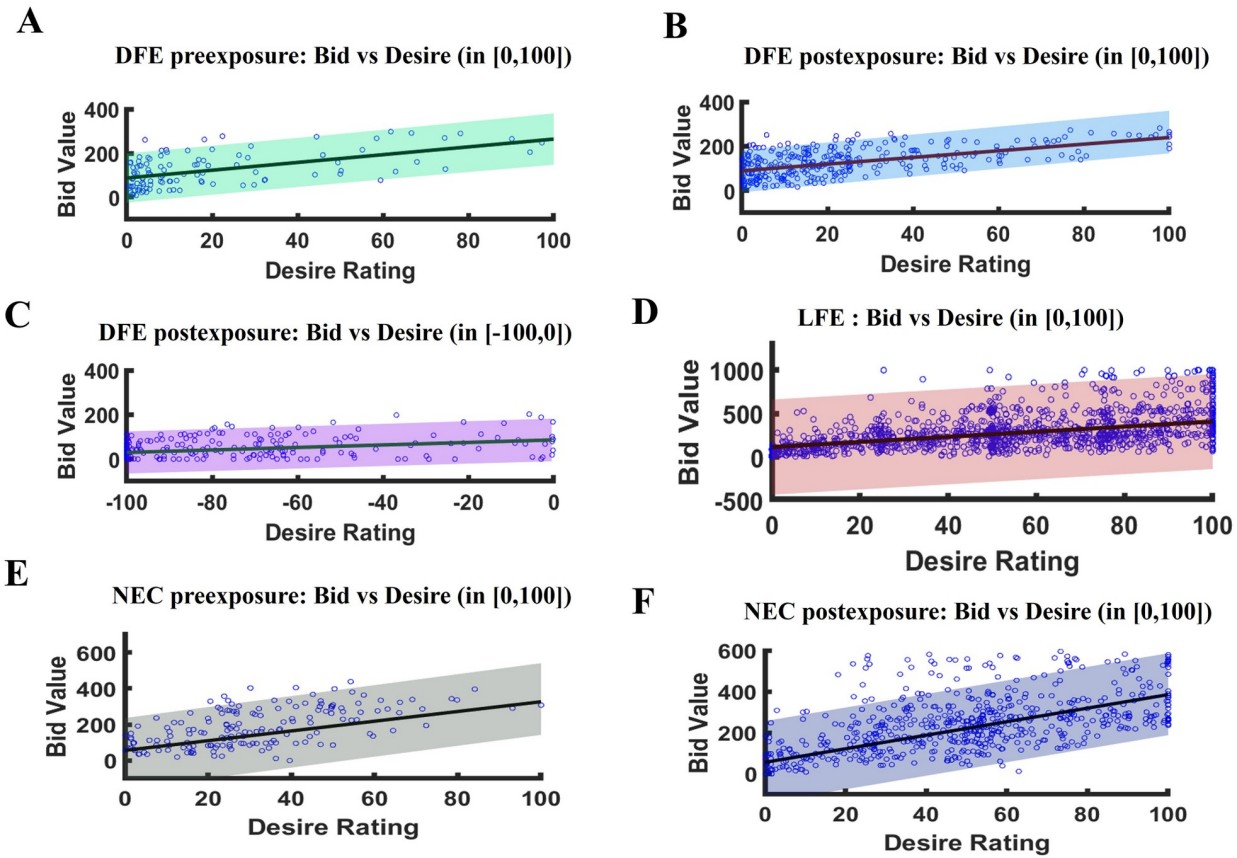

**Fig 6. Model prediction of bid value from desire rating.** A: DFE preexposure model when the desire rating is within the interval [0, 10]. B: DFE postexposure model when the desire rating is within the interval [0, 10]. C: DFE postexposure model when the desire rating is within the interval [-100,0). D: LFE model when the desire rating is within the interval [0, 10]. E: NEC model at preexposure. F: NEC model at postexposure. Blue dots are the data points and the straight line along with the colored shaded region indicate SRM prediction with 95% confidence intervals. Almost all the data points are within the 95% confidence interval, which shows the SRM fits the data well for all participants' categories.

significant ($F_{5,1082} = 1065$, $P < 2.2 \times 10^{-16}$, Adjusted R—squared = 0.83) and all but $\gamma_1$ (coefficient of desire in [-100,0)) and $\gamma_4$ (coefficient of reaction time) were significant (see Table 4). Here also, like in disliked food exposure studies, we have checked the predictive performance of our model by estimating the 95% predictive band. The band along with the true values are

**Table 3. Spline regression model results.** $\beta_0$ and $\beta_2$ are the coefficients of the intercepts when desire value is in [-100,0) and [0, 10] respectively. $\beta_1$ and $\beta_3$ are the coefficients of the desire value in [-100,0) and [0, 10] respectively. $\beta_4$ is the coefficient of reaction time. Coefficients with highlighted p-values are significant in the model. $^*p < 0.05$, $^{**}p < 0.01$, $^{***}p < 0.001$.

| Spline regression model (DFE preexposure). | | | |
|---|---|---|---|
| Coefficients | Estimate | t-value | p-value |
| $\beta_0$ | 86.4553 | 8.693 | $<2 \times 10^{-16}$ *** |
| $\beta_1$ | 0.5655 | 5.43 | $8.42 \times 10^{-8}$ *** |
| $\beta_2$ | 89.8364 | 15.906 | $<2 \times 10^{-16}$ *** |
| $\beta_3$ | 1.4947 | 13.273 | $<2 \times 10^{-16}$ *** |
| $\beta_4$ | 0.001734 | 1.004 | 0.316 |

**Table 4. Spline regression model results.** $\gamma_0$ and $\gamma_2$ are the coefficients of the intercepts when desire value is in [-100,0) and [0, 10] respectively. $\gamma_1$ and $\gamma_3$ are the coefficients of the desire value in [-100,0) and [0, 10] respectively. $\gamma_4$ represents the coefficient of reaction time. Coefficients with highlighted p-values are significant in the model. $^*p < 0.05$, $^{**}p < 0.01$, $^{***}p < 0.001$.

| | Spline regression model (LFE). | | |
|---|---|---|---|
| **Coefficients** | **Estimate** | **t-value** | **p-value** |
| $\gamma_0$ | 55.63 | 2.062 | 0.0394 $^*$ |
| $\gamma_1$ | −0.0699 | −0.041 | 0.9672 |
| $\gamma_2$ | 113.9 | 10.823 | $<2 \times 10^{-16}$ $^{***}$ |
| $\gamma_3$ | 2.95 | 21.751 | $<2 \times 10^{-16}$ $^{***}$ |
| $\gamma_4$ | −0.0003 | −0.098 | 0.9218 |

displayed in Fig 6D, which clearly showed the band captures most of the true values. This indicates that our proposed model is not suffering from overfitting.

In the LFE model, the non-significant $\gamma_1$ showed no effect of desire value on bid value in the negative desire range, and reaction time shows no effect in the model. However, the model depicted a significant effect of desire value on bid value in the non-negative desire range.

**For NEC.** Change point detection analysis confirmed no significant change in mean bid-value at preexposure compared to the postexposure blocks but a significant change in mean desire value at preexposure compared to the postexposure blocks for the exposed series. Thus, we have considered a spline regression model, with a knot at the fifteenth trial for the entire exposed series. The proposed NEC model (4) can be written as follows:

$$BV = \delta_0 + (\delta_1 DV)I(t \leq 15)) + (\delta_2 DV)I(t > 15)) + \delta_3 RT + \epsilon, \tag{4}$$

where $BV$, $DV$, $RT$ & $t$ denote the bid value, desire value, reaction time of bid value, and trial number respectively, for the NEC group and $\delta$'s are the coefficients of the model. This model turned out to be significant ($F_{3,1185} = 581.9$, $P < 2.2 \times 10^{-16}$, Adjusted R—squared = 0.59) and all the coefficients were significant (see Table 5) in the model. To nullify the possibility of overfitting, like in disliked food exposure and liked food exposure studies, the predictive bands for preexposure and postexposure models were estimated. The 95% bands and the true values are depicted in Fig 6E and 6F. As expected, the major portion of the true values fell within the bands.

The NEC model depicts a significant effect of desire value on bid value for the exposed food items throughout the experiment and the association between bid value and desire value increased slightly after multisensory food exposure (since $\hat{\delta}_2 - \hat{\delta}_1 = 0.5873$).

Thus, our computational models clearly showed that for exposed food items, willingness to pay (measured by bid value) could be predicted successfully from desire values for all

**Table 5. Spline regression model results.** $\delta_0$ represents the coefficient of the intercept. $\delta_1$ and $\delta_2$ are the coefficients of the desire value in preexposure and postexposure respectively. $\delta_3$ represents the coefficient of the reaction time. Coefficients with highlighted p-values are significant in the model. $^*p < 0.05$, $^{**}p < 0.01$, $^{***}p < 0.001$.

| | Spline regression model (NES). | | |
|---|---|---|---|
| **Coefficients** | **Estimate** | **t-value** | **p-value** |
| $\delta_0$ | 26.6446 | 5.506 | $4.5 \times 10^{-8}$ $^{***}$ |
| $\delta_1$ | 2.7018 | 13.084 | $<2 \times 10^{-16}$ $^{***}$ |
| $\delta_2$ | 3.2891 | 35.381 | $<2 \times 10^{-16}$ $^{***}$ |
| $\delta_3$ | 0.0198 | 11.009 | $<2 \times 10^{-16}$ $^{***}$ |

categories of participants. In particular, for DFE, our model showed an insignificant preference for disliked food prior to exposure, which was overturned, and became significant following multisensory exposure. Our results indicated a possible reversal effect of food preference after multisensory food exposure for DFE participants. In NEC participants, momentary desire for food items did not translate to willingness to pay alluding to the importance of momentary valuation of displayed food cues in craving studies.

## Univariate EEG analysis

To determine the timing and location of the maximum amplitudes, we have observed the topographic plots of grand averages at different time points for both preexposure and postexposure conditions using EEGLAB Toolbox [48]. Further, to elucidate whether the preexposure and postexposure conditions induce different neural processing mechanisms, the grand average difference waveform was observed. A clear difference in preexposure and postexposure ERPs (N200 at 200ms and P300 at 300 ms) was visible only for DFE (see Fig 7).

**P200.** We have observed a positive going peak around 200 ms after stimulus onset for all participant categories (see Fig 8A–8C). However, there was no significant mean difference between preexposure and postexposure for any of the participant categories (DFE: $t(11) = 0.7636$, $p = 0.2306$, $d = 0.22$, $BF = 0.5555$, LFE: $t(11) = -0.4504$, $p = 0.6694$, $d = 0.13$, $BF = 0.2142$ and NEC: $t(10) = 0.0538$, $p = 0.4791$, $d = 0.02$, $BF = 0.3094$). Further, all participant categories showed no localization effect (left vs right) of P200 amplitude at preexposure and postexposure (see S3 Table in S1 File for details).

**Posterior N200.** N200 peak was observed around 200 ms after stimulus onset for all participant categories (see Fig 8D–8F). No significant mean difference between preexposure and postexposure conditions was observed for the LFE and NEC participants (LFE: $t(11) = -0.8296$, $p = 0.7878$, $d = 0.24$, $BF = 0.1756$ and NEC: $t(10) = -1.1249$, $p = 0.8565$, $d = 0.34$, $BF = 0.1609$) but the exposed food items for DFE at postexposure (M = $-3.88\mu V$, SD = 2.54) showed significantly higher negative amplitudes (DFE: $t(11) = 2.3074$, $p = 0.0207$, $d = 0.67$, $BF = 3.7173$) than the preexposure exposed food items (M = $-2.86\mu V$, SD = 3.15, see Fig 8D). We have also observed a localization effect (left vs right) at both preexposure and postexposure only for the NEC group (see S4 Table in S1 File for details).

**P300.** We have observed a positive going peak at the centro-parietal electrode cluster around 300 ms after stimulus onset for all participant categories (see Fig 8G–8I). Among all the participant categories, no significant mean difference between preexposure and postexposure conditions was observed for the LFE and NEC participants (LFE: $t(11) = 0.4157$, $p = 0.3428$, $d = 0.12$, $BF = 0.4008$ and NEC: $t(10) = 0.2288$, $p = 0.4118$, $d = 0.07$, $BF = 0.3542$). However, the DFE category showed a more positive amplitude ($t_{11} = 2.2523$, $p = 0.0229$, $d = 0.65$, $BF = 3.4416$) in response to exposed food pictures at preexposure condition (M = $3.60\mu V$, SD = 2.68) as compared to the exposed food pictures at postexposure condition (M = $2.76\mu V$, SD = 2.66, see Fig 8G). We have also observed a localization effect (left vs right) at both preexposure and postexposure for the NEC group only (see S5 Table in S1 File for details).

**Late positive potential.** A sustained positive amplitude from 300 ms to 600 ms at poststimulus onset was observed for all participant categories (see Fig 8G–8I). However, there was no significant mean difference between preexposure and postexposure conditions for any of the participant categories (DFE: $t(11) = 0.3096$, $p = 0.3813$, $d = 0.09$, $BF = 0.3660$, LFE: $t(11) = 0.6888$, $p = 0.2526$, $d = 0.20$, $BF = 0.5159$ and NEC: $t(10) = -0.1352$, $p = 0.5524$, $d = 0.04$, $BF = 0.2706$). Further, We have found a localization effect (left vs right) at postexposure conditions for LFE and NEC (see S6 Table in S1 File for details).

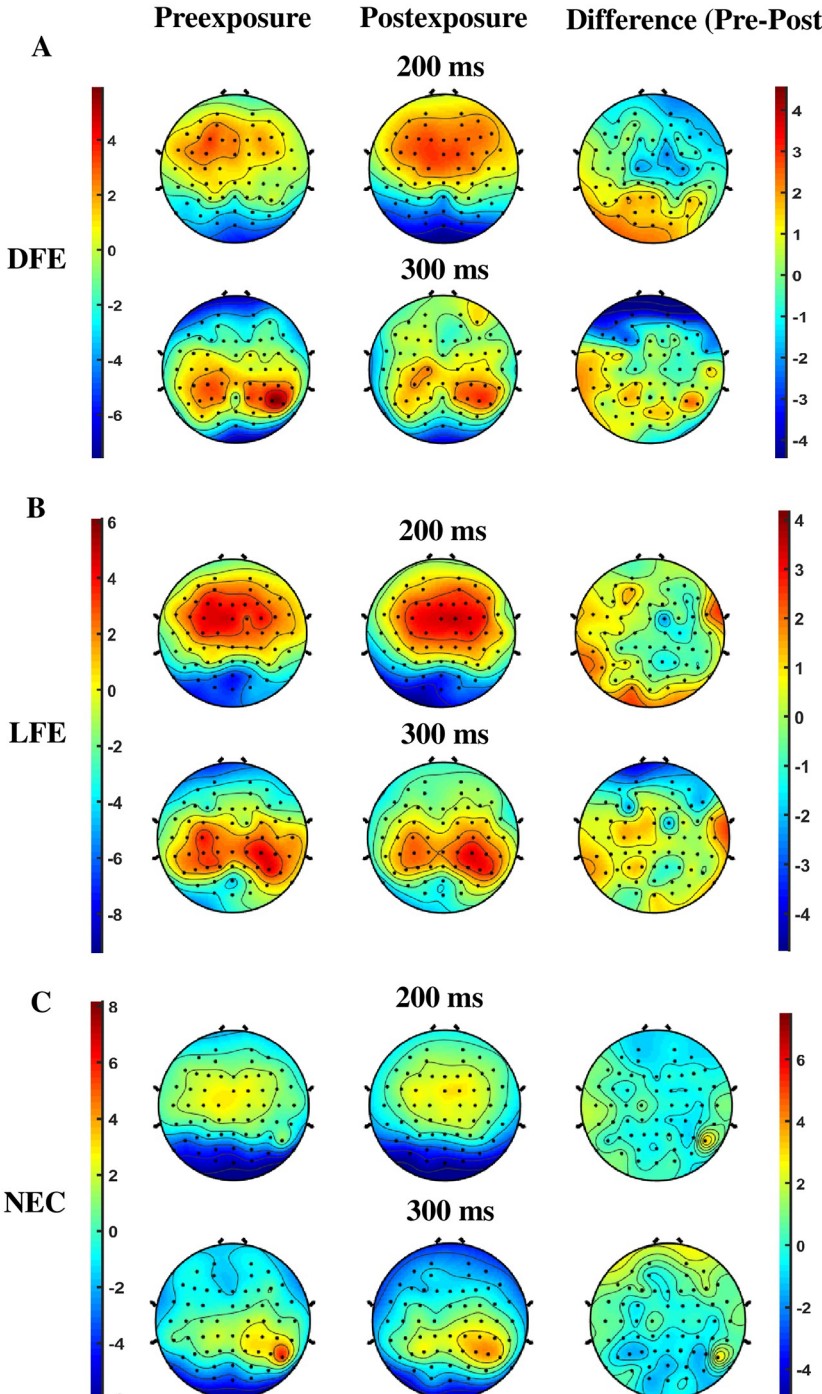

**Fig 7. Topoplots for preexposure, postexposure and the their differences (preexposure-postexposure).** Topoplots are depicted for A: DFE, B: LFE and C: NEC. at 200ms and 300ms. The colorbars at left side are for the reference of both preexposure and postexposure, whereas colorbars at right side are for the reference for the differences.

## Correlation analyses

Partial correlation matrices of subjective desire ratings and electrophysiological dependent variables were computed for all participant categories (see S7 Table in S1 File for details). DFE

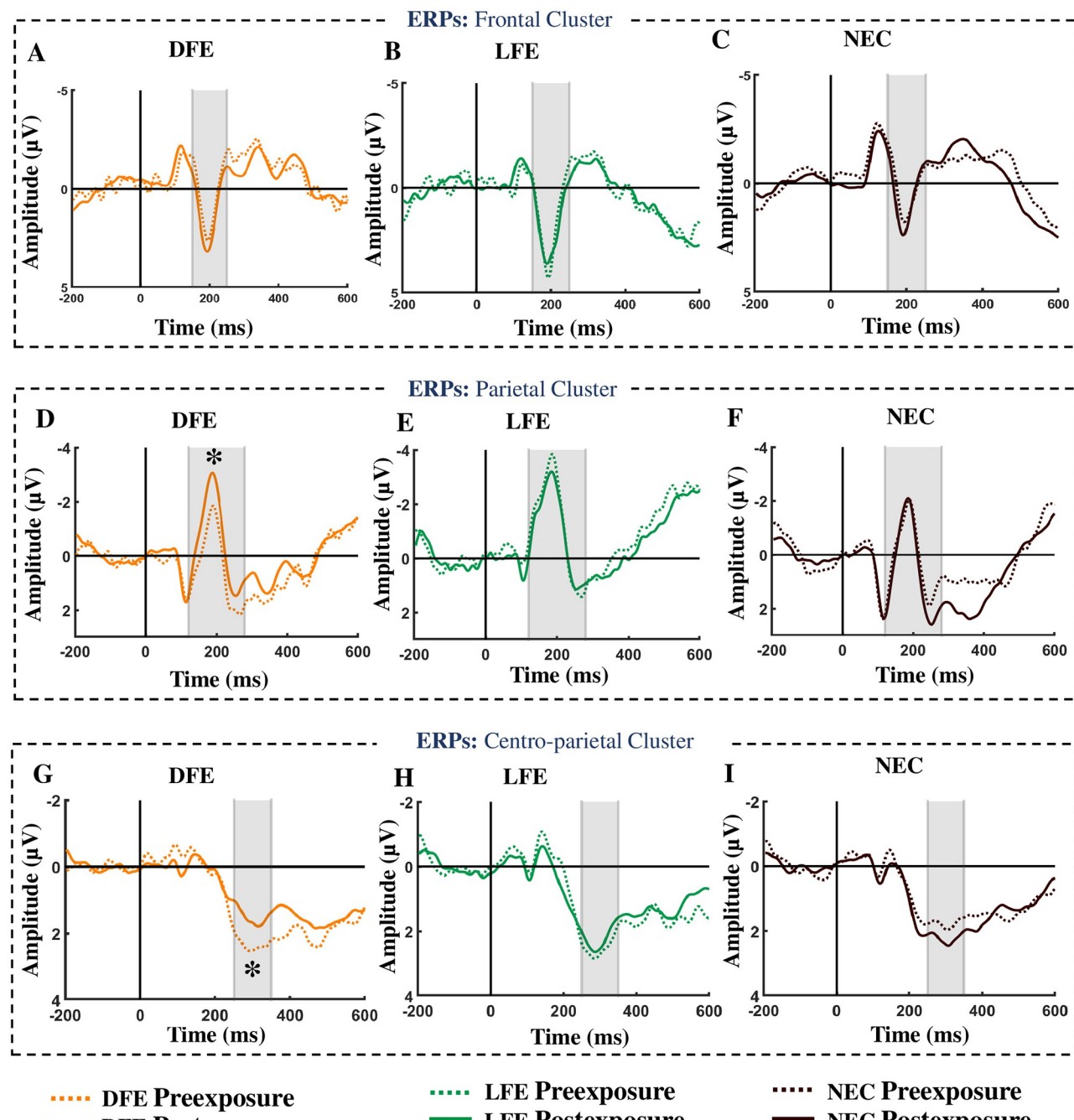

**Fig 8. Event-related potentials in response to the food pictures (exposed) of a frontal cluster (F1, F2, F3, F4, Fz, AF3, AF4), parietal cluster (P1, P2, P3, P4, Pz, PO3, PO4) and centro-parietal cluster (CP1, CP2, CP3, CP4, CPz, P1 and P2) for both preexposure and postexposure conditions.** The figure depicts the electrophysiological activity from 200 ms before to 600 ms after stimulus onset. The subplots A, B and C depict the ERPs from the frontal cluster for DFE, LFE, and NEC participants, respectively. The grey-shaded regions of A, B, and C indicate the time window when the positive going peak is identified. There seems to be no difference in P200 amplitude between preexposure and postexposure for all the participant categories. The subplots D, E, and F depict the ERPs from the parietal cluster for DFE, LFE, and NEC participants, respectively. The grey-shaded regions of D, E, and F indicate the time window when the negative going peak is identified. There seems to be a clear difference in N200 amplitude between preexposure and postexposure only for DFE. The subplots G, H, and I depict the ERPs from the centro-parietal cluster for DFE, LFE, and NEC participants, respectively. The grey shaded region of G, H, and I indicate the time window when the positive going peak is identified. There seems to be a clear difference in P300 amplitude between preexposure and postexposure only for DFE.

and LFE participants showed moderate to very high correlations between desire and ERP components except in P200, in both preexposure and postexposure conditions. However, the post-exposure condition produces stronger correlations as compared to preexposure for both DFE and LFE. We found a low correlation in the case of NEC desire and ERPs.

### Single-trial multivariate analysis

A pattern classifier (CPCA) [58] was used to analyze single-trial EEG signals corresponding to the preexposure and postexposure conditions for all participant categories. The classifier was trained using the EEG signals from the frontal, parietal, and centro-parietal electrode clusters (see Fig 9B) spanning the corresponding time interval of the ERP components. Leave one trial out cross validation method for each of the individual participants was performed and mean classification accuracy for each of the participants was noted. The overall mean accuracy for all participants along with standard error is shown in Fig 9A. Only DFE participants show above chance classification accuracy (DFE: 0.56, $t_{11} = 3.0913$, $p = 0.0051$, LFE: 0.47, $t_{11} = -0.8804$, $p = 0.80$, NEC: 0.53, $t_{10} = 1.4072$, $p = 0.09$, Fig 9).

Our neural analyses point to the role of posterior N200 and centroparietal P300 in modulating the neural response between pre and post food exposure for DFE participants. The univariate results are corroborated by single trial multivariate pattern analyses.

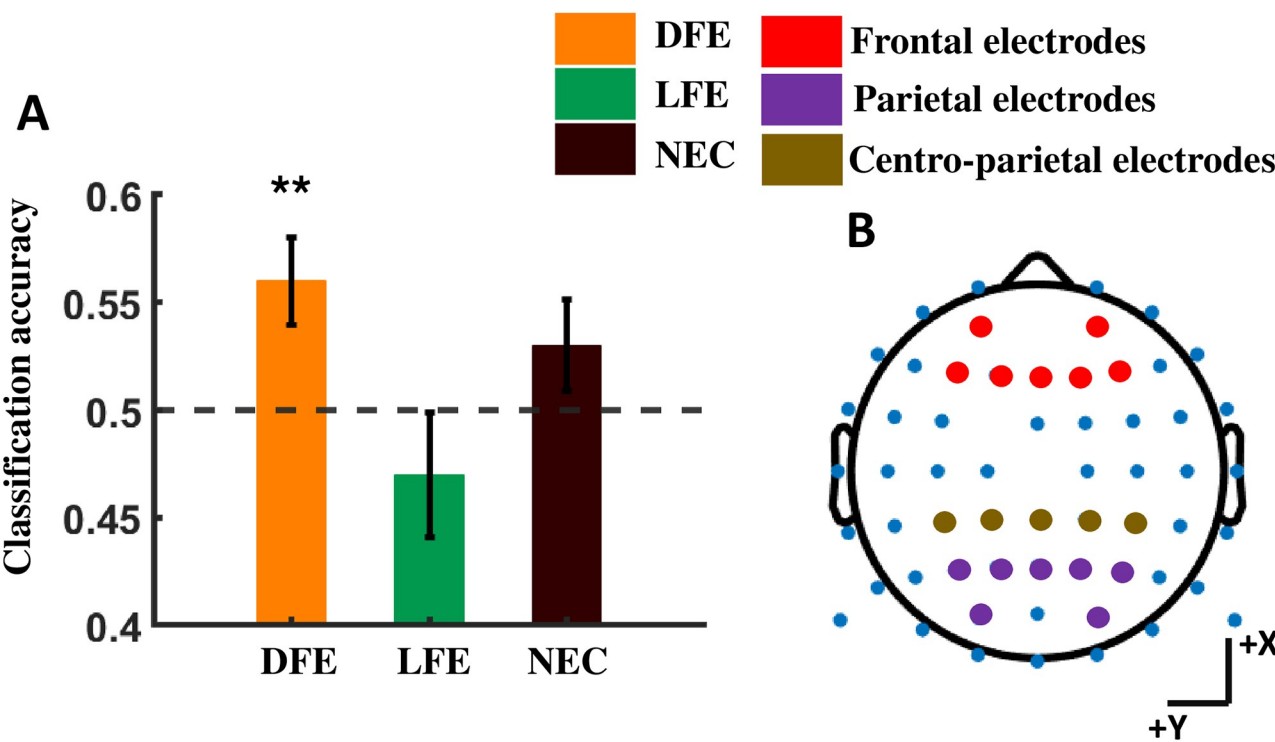

**Fig 9. Classification accuracy between preexposure and postexposure for all participant categories.** A. The orange, green and brown bars represent mean classification accuracies for DFE, LFE and NEC respectively. The horizontal dashed line represents the chance classification. B. Frontal (F1, F2, F3, F4, Fz, AF3, AF4), centro-parietal (CP1, CP2, CP3, CP4, CPz) and parietal (P1, P2, P3, Pz, PO3, PO4) electrodes on a 64-channel EEG cap. There seems to be an above chance classification accuracy for the DFE only. $^*p < 0.05$, $^{**}p < 0.01$, $^{***}p < 0.001$.

## Discussion

Rising obesity and craving related disorders including binge eating disorder necessitate a detailed understanding of the neuro-cognitive mechanism behind food craving. Despite some recent work [6, 23], our understanding of food craving remains limited. In this study, we explored craving through computational modeling and statistical analysis by subjecting participants to food cues and multisensory food exposure. We grouped the participants based on their choice of liked the food (sweet item or savory item) and systematically exposed them to their liked or disliked food item. They were grouped as Liked food exposure, Disliked food exposure, or Neutral depending on their liking for exposed food items. We explored the effect of multisensory exposure on liked and disliked food items and computed a statistical model to predict the subjective value the participants were willing to pay to obtain the craved item. Using univariate and multivariate neural data processing, we studied the neural markers and timeline of food craving. The goal of the study was to explore the effect of food craving pre and post multisensory exposure within each group of participants. Our objective is not to compare any between group effect, hence, throughout our study, we do not consider any $n$-way ANOVA ($n > 1$) to compare between groups of the participants, i.e., DFE, LFE, and NEC.

### Inducing craving for disliked food items

Accessibility to inexpensive unhealthy food combined with aggressive marketing has led to a substantial rise in craving related disorders in the last decade. Thus there is an urgent need to develop efficient intervention strategies to promote healthy eating. Public policy interventions include strategies like reducing availability or increasing the price of unhealthy food [16]. Intervention studies have tried to address this issue by using behavioral [16, 31], financial [65] and cognitive strategies [6, 9–12]. One approach in these intervention studies includes nudging people to healthier food choices mostly using priming and saliency [66], although the efficacy of such studies is mixed. Most of the intervention studies focus on down-regulating craving using cognitive strategies and had consistently shown that delaying gratification [3, 11], thinking about negative long term consequences reduce craving for both food [5, 6] and drugs [6, 9]. Our results show that it is possible to up-regulate desire and WTP for previously disliked food items by exposing participants to disliked food. By grouping participants based on their liking of particular food items, we were able to systematically observe the effect of food cues and food exposure on liked and disliked food items. Our results establish that multisensory food exposure was instrumental in inducing craving irrespective of likeness to a particular food item. Thus, for participants disliking a particular food item (DFE participants), it is possible to induce liking for disliked food by subjecting them to multisensory food exposure. Our results also show a significant reduction in desire values for previously liked food items for DFE participants following exposure to disliked food thus showing a potential reversal of food preference. Reversal effect for disliked food also translated to subjective values of food items and willingness to pay. Our results seem to imply the feasibility of using multisensory exposure to reverse food preference and it could potentially contribute to designing better intervention studies and have future policy implications.

### Role of food exposure and willingness to pay in craving

Food craving studies generally show that the desire for exposed or displayed food items increases the momentary desire and subjective value of the food item. It has been previously demonstrated that the valuation of food is influenced by the format used for food display [67] and showing real food garnered more response compared to 2d images of food [67, 68]. However, majority of studies exploring food choices and craving use food cues and mental imagery

in the experimental paradigm, and to the best of our knowledge, only a couple of studies exist [23, 24] where multisensory exposure was included in the experimental paradigm. In this study, we demonstrated that multisensory exposure to food items produces a significant effect on craving as compared to food cues. Our behavioral analyses showed that postexposure desire and bidding ratings were always higher for the food item and this effect is significant in DFE participants (see Fig 4). FCQ-S scores pre and post experiment showed that craving was induced for all three categories of participants (liked food exposure, disliked food exposure, and neutral control) significantly post experiment (see Fig 3). Our results also showed that there is no time dependency between the inducement of craving and the last meal taken by participants (see Table 1). Hence for craving-related research, it is not necessary to bar participants from having food for a long time, a fact often ignored by most studies [23, 69].

In majority of the studies [24, 25], the participant's response is typically given in terms of desire rating for the displayed food cues which does not affect the valuation of displayed food. However, whether the desire for a particular food item translates to WTP often remains unexplored [23]. Our results show that an increase in desire does not always lead to an increase in WTP. In the case of neutral control participants, multiple analyses showed that although the desire for exposed food items increased momentarily, it did not translate to a significant increase in concurrent subjective valuation of food items which was measured by the bidding scale when compared with preexposure bidding data. This is an important result from the perspective of consumer behavior and establishes the importance of recording subjective valuation of food cues.

## Neural mechanism of craving

Cognitive functioning plays a role in modulating food craving and can help elucidate the underlying neural mechanism. Neuroimaging studies have revealed that food cues inducing craving leads to significant neural activity in the ventral striatum (VS) [14, 70], amygdala [71, 72], insula [72, 73], ventral tegmental area (VTA) [14, 71], orbitofrontal cortex [14, 71, 72, 74] and anterior cingulate cortex (ACC) [14, 71, 74]. Most of these regions are known to modulate emotion, and motivation and are part of the dopamine reward pathway [75, 76]. Similar results are also found in drug addiction studies [77–79].

The neural timeline modulating food craving could be reflected by ERP components measured during the experimental paradigm. Initial sensory and attention allocation towards food cues are reflected in increased P200, N200 components [30, 35] as compared to neutral stimuli. Higher-order attentional allocation was reflected in increased P300 and LPP for more palatable food [12], neutral stimuli [80] or less emotionally charged food [81, 82].

We have shown using behavioral analyses and modeling that participants disliking a particular food item (DFE participants) on exposure to the food reverses her/his previous preference and develops desire for exposed food. Our behavioral results are also corroborated using multivariate pattern analysis on neural signals showing a significant difference between preexposure and postexposure for DFE participants. Unlike most previous studies, in the current setup, participants are exposed to food cues throughout the experiment and we compare the ERP amplitudes pre and post multisensory exposure to food. Our results show that there is no significant difference between participant categories for P200 and LPP whereas, for the DFE participants, N200 and P300 are significantly different pre and post food exposure. An increase in parietal N200 and a decrease in centroparietal P300 postexposure of disliked food was observed for DFE participants.

We hypothesize that an increase in parietal N200 and a decrease in P300 following food exposure possibly modulate the craving mechanism and result in an increased desire and

valuation for previously disliked food items. Increase in N200 has been previously associated with high calorie food [80], initial sensory or visual attention (see [35] for review). An increase in liking for disliked food in DFE participants following food exposure is seen to be associated with an increase in N200, possibly signifying an increase in the initial allocation of attention and decreased inhibition to a previously disliked food item.

It is interesting to note that we get a significant difference in the P300 components which is typically found in food-related inhibitory control and intervention studies [38, 83, 84]. In intervention studies, a decrease in P300 has been related to cognitive regulation strategies [10] and short-term consequences for consuming high-calorie food [11]. P300 is a known ERP marker for response inhibition [85] and while frontal N200 can reflect initial inhibition processing, P300 can account for increased recruitment for inhibitory control processes [86, 87]. A recent craving study advocates the role of only P300 and not N200 in inhibitory control deficits in obese people. In the study by Kong et al. [80], an increase in P300 for neutral and low-calorie food compared to high-calorie food for successful restrained eaters was shown pointing to the role of larger P300 amplitude for increased recruitment of inhibitory control process. In the current setup, preexposure conditions for disliked food exposure participants produce a similar scenario as seen in inhibitory control studies leading to an increase in the P300 component for disliked food. However, following food exposure, the preference for exposed food for disliked food exposure participants is seen to be reversed which is possibly reflected in the lowering of inhibitory control and hence associated with a decrease in P300 amplitude.

## Limitations and future work

We have demonstrated that with food exposure, craving can be induced for disliked food items however both disliked and liked food items consisted of high-calorie food in the current study. The generalization of the current experimental paradigm can be strengthened by follow-up studies using similar paradigms consisting of both high calorie and low calorie food. We have not controlled for the number of participants having sweet or savory items as exposed food. Since the type of food seems to affect craving tendencies, a detailed study controlling for different food types could further elucidate our understanding of the craving mechanism. In the current study, the state of satiety and hunger assessment before beginning the experiment was not considered. Only time refrained from eating was accounted for but the type of food consumed and state of hunger before the experiment were not considered. The effect of satiety and hunger on the craving for liked and disliked food can be considered in future studies. Similarly, the effect of craving following multisensory exposure on participants on diet and/or suffering from eating-related disorders could also enhance our existing knowledge of food craving.

## Conclusion

The current study aims at manipulating food craving preferences using multisensory food exposure. Grouping participants based on their food preferences, the study shows that craving can be induced through food exposure even for disliked food items. Neural markers modulating craving are explored and the results seem to point to the role of reduction of inhibitory control for reversing momentary food preference. Further studies in modulating food craving following similar paradigms could potentially have a long-term impact on public health policy.

## Supporting information

**S1 File.**
(PDF)

## Acknowledgments

The authors are thankful to the anonymous reviewer for her/his insightful comments, which improves the earlier version of manuscript significantly.

## Author Contributions

**Conceptualization:** Satyaki Mazumder, Koel Das.

**Data curation:** Avishek Chatterjee.

**Formal analysis:** Avishek Chatterjee.

**Funding acquisition:** Koel Das.

**Investigation:** Avishek Chatterjee.

**Methodology:** Satyaki Mazumder, Koel Das.

**Project administration:** Satyaki Mazumder.

**Resources:** Avishek Chatterjee.

**Software:** Avishek Chatterjee.

**Supervision:** Satyaki Mazumder, Koel Das.

**Validation:** Avishek Chatterjee, Satyaki Mazumder.

**Visualization:** Avishek Chatterjee.

**Writing – original draft:** Avishek Chatterjee.

**Writing – review & editing:** Satyaki Mazumder, Koel Das.

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
