## [Decision Letter · Decision Letter 0]

24 Apr 2023

PONE-D-22-32995Reversing Food Preference Through Multisensory ExposurePLOS ONE

Dear Dr. Das,

Thank you for submitting your manuscript to PLOS ONE. After careful consideration, we feel that it has merit but does not fully meet PLOS ONE’s publication criteria as it currently stands. Therefore, we invite you to submit a revised version of the manuscript that addresses the points raised during the review process.

We look forward to receiving your revised manuscript.

Kind regards,

Joydeep Bhattacharya

Academic Editor

PLOS ONE

Journal Requirements:

5. We note that Figures 1 and 2 in your submission contain copyrighted images. All PLOS content is published under the Creative Commons Attribution License (CC BY 4.0), which means that the manuscript, images, and Supporting Information files will be freely available online, and any third party is permitted to access, download, copy, distribute, and use these materials in any way, even commercially, with proper attribution. For more information, see our copyright guidelines: http://journals.plos.org/plosone/s/licenses-and-copyright.

a. You may seek permission from the original copyright holder of Figures 1 and 2 to publish the content specifically under the CC BY 4.0 license. 

Additional Editor Comments:

First off, I am really sorry for the inordinate delay in getting to a decision. For some reason, it was incredibly difficult to find suitable referees (more than a dozen people were approached). I have now received one solid report from a knowledgeable referee, and based on their report, I am happy to offer you a chance to revise and submit your paper. It is a major revision. The referee has mixed feelings. They think the design introduces confounds and disturbs the clean identification. To be acceptable for publication, you would need to significantly clarify and provide more rationale and background on the Becker-DeGroot-Marschak mechanism of the task (i.e. provide more detail and justification on the modification they mention making to it), potentially reinterpret results and conclusions as appropriate, and both mention and elaborate upon this in the study limitations in your discussion. I'd also like to see a point-by-point discussion of the referee's main comments. I will take a final decision in the next round. No promises, of course, at this stage.

Reviewers' comments:

Reviewer's Responses to Questions

**Comments to the Author**

1. Is the manuscript technically sound, and do the data support the conclusions?

Reviewer #1: Yes

2. Has the statistical analysis been performed appropriately and rigorously? 

Reviewer #1: Yes

3. Have the authors made all data underlying the findings in their manuscript fully available?

Reviewer #1: No

4. Is the manuscript presented in an intelligible fashion and written in standard English?

Reviewer #1: No

5. Review Comments to the Author

Reviewer #1: Reviewer Comments for the Authors

INTRODUCTION

Overall comments:

1. The authors could better summarize past literature on ERPs by highlighting and describing the work done in a few select seminal papers demonstrating N200/N2, P2, P300, and LPP’s respective importance and association with attention, food stimuli, and inhibitory control, which would provide more context for the chocolate stimuli they reference.

2. The manuscript’s introduction could be more clearly laid out by having the authors group all their specific aims for the manuscript in a section at the end of the introduction instead of presenting them throughout the introduction. This would provide a better segue from their review of the literature, regarding what is known, to how they now approach filling in these research gaps with their specific aims for the current study.

3. Following the above comment, in some instances the aims of the paper seem to be repeated. It would be helpful if the authors could more succinctly summarize their aims and objectives perhaps grouping them as:

Whether behavioral desire ratings for craved foods translates to Willingness-To-Pay value measures

Prediction of Willingness-To-Pay value for craved food items

Efficacy of the Reversal Food Preference for the food craving induction for disliked foods

Neural correlates of food craving and its modulation (univariate and multivariate analyses)

4. In addition to two previous points made above, the methods and techniques used to achieve each of these aims and objectives are sporadically referenced throughout the introduction as the aims are not consolidated in one specific section. By marrying each brief description of the approach used with each aim/objective the authors would be able to improve the clarity of their manuscript and better layout the experiments they detail in the Methods and Results sections of the manuscript.

5. The reversal of food preference is mentioned only at the end of the introduction. As it is highlighted in the title of the manuscript and a main topic of the manuscript, the authors should consider mentioning it sooner in their introduction and place more emphasis on its importance and relevance in their study, perhaps introducing it along with or after their summary on past literature on the regulation of craving.

Specific comments:

6. The wording, “progress from craving to spending,” on page 3 could be clarified, perhaps by using the word valuation instead. Here, the authors could also emphasize the progression and motivation behind acting on cravings, expanding on this idea as well.

7. Similarly, the wording, “efficient public health policies” on page 3 could be improved by saying replacing “efficient” with “effective”.

MATERIALS & METHODS

Overall Comments:

8. The authors could describe the criteria for their sample better by providing both an inclusion and exclusion criteria for their participants.

9. The methodology and experimental timeline of the task are intermixed in the Methods section, making it hard to follow how the experiment was carried out. The authors could improve clarity on this by detailing their methods and measures used in the study prior to then separately outlining the timeline of the task in another section.

10. The labels “Experiment 1” and “Experiment 2” that the authors used would benefit from having more descriptive names as Experiment 1 has no EEG component and is primarily behavioral, while Experiment 2 includes EEG with the task. The authors should consider renaming “Experiment 1” and “Experiment 2” to “Behavioral Experiment” and “EEG Experiment”, respectively.

11. It would be helpful if the authors specified what modification they made to the Becker-DeGroot-Marshak task when they discuss it on page 6 and include references to past literature validating this approach.

12. Discussion of the goals of the study in the methods section should be omitted and moved into the introduction and discussion as appropriate.

Specific Comments:

13. The authors should explicitly state what variables were used in their Spline Regression Models for clarity on page 9.

14. Sample sizes should be reported with (n=#) notation throughout the manuscript.

15. The authors should consider spelling out rather than abbreviating “viz.” on page 8.

16. The authors should include Table 1 in the Results section rather than the Materials and Methods section.

RESULTS

Overall comments

17. The authors should consider restructuring the results section by summarizing and comparing the results for all the different groups (DFE, LFE, NEC) for each of the analyses together instead of reporting the results by group in three different sections. This would cut down on repeated explanations of what the analysis is and what was done and relieve the need to have a summary at the end of the Results section as the main findings between the group comparisons are summarized in the respective analysis sections previously.

18. The authors should also consider including change point detection plots for the non-exposed foods. This would help visualize the magnitude changes in bid and desire ratings for the food item used in the multisensory food exposure experience relative to the food items participants were not exposed to.

Specific Comments:

19. For clarity the authors should more explicitly state which Chi-square test is for their behavioral experimental sample and which is for the EEG experimental sample on page 10.

20. The authors should consider adding more descriptive title and axis labels to Figure 3 to distinguish the behavioral and EEG panels of the figure.

21. Additionally for each panel in Figure 3, the group comparison of FCQ-S scores would be easier to read and visually compare if the before and after bars for each group (DFE, LFE, NEC) were shown side-by-side.

22. Figure 4 would benefit from using different hues of the experimental group’s color to distinguish between the bid value and desire responses plotted for each group. Additionally it would be good to add visualizations of the statistical tests done to the plots to better highlight the change point findings reported in the results.

23. The authors should consider refraining from describing models with superlatives like “best model” on page 12, perhaps using words like “optimal” instead.

24. The authors should consider adding axis labels and subplot titles in Figure 5 to clarify which groups what pre- or post-exposure model being shown in each panel. Additionally the authors should consider using a maker other than an asterisk in these plots, such as open circles, so the density of points can be better assessed as the asterisks are harder to distinguish from one another when they overlap.

25. Figure 7, 8, and 9 could be collapsed into one figure with the same legend, with separate panels for each of the different topological clusters EEG data was recorded from.

DISCUSSION

Overall comments:

26. The authors should be cautious when using the word craving throughout their discussion. Their results suggest that individuals do enter a state of craving, but from what they describe in their methodology and the results they present it is unclear if this food craving is specifically for the disliked food item or if it is that they are experiencing some form of a craving.

Specific Comments:

27. The authors are unclear in part of their statement on page 21 that states, “... increase in desire does not always lead to WTP.” The authors should clarify whether they mean an increase or decrease of WTP.

6. PLOS authors have the option to publish the peer review history of their article (what does this mean?). If published, this will include your full peer review and any attached files.

Reviewer #1: No

---

## [Author Response · Author response to Decision Letter 0]

21 Jun 2023

Additional Editor Comments:

First off, I am really sorry for the inordinate delay in getting to a decision. For some reason, it was incredibly difficult to find suitable referees (more than a dozen people were approached). I have now received one solid report from a knowledgeable referee, and based on their report, I am happy to offer you a chance to revise and submit your paper. It is a major revision. The referee has mixed feelings. They think the design introduces confounds and disturbs the clean identification. To be acceptable for publication, you would need to significantly clarify and provide more rationale and background on the Becker-DeGroot-Marschak mechanism of the task (i.e. provide more detail and justification on the modification they mention making to it), potentially reinterpret results and conclusions as appropriate, and both mention and elaborate upon this in the study limitations in your discussion. I'd also like to see a point-by-point discussion of the referee's main comments. I will take a final decision in the next round. No promises, of course, at this stage.

We thank the editor for giving us a chance to address the reviewer's comments. We have modified the manuscript significantly as per the reviewer's suggestions and address all the comments to the best of our abilities. Regarding Becker-DeGroot-Marschak mechanism, I would like to clarify that we did not modify the BDM mechanism and only used it differently than the way it was used in a previous study (Konova et al, 2018). The details were given in the supplementary information but the way it was mentioned in the main text was misleading. We thank the reviewer and editor for bringing it to our notice and we have modified the text in the manuscript and it now reads 

"Participants received Rs, 100 and a snack food item that he/she desires the most for their participation and a bonus amount, which was calculated using standard Becker-DeGroot-Marschak (BDM) mechanism [47]. Instead of considering BDM on a single trial from the last completed block [23], we randomly selected one out of 13 blocks and considered all trials in that block for the bonus calculation using standard BDM auction (see Supporting Information for details". 

Reviewers' comments:

Reviewer's Responses to Questions

Comments to the Author

1. Is the manuscript technically sound, and do the data support the conclusions?

Reviewer #1: Yes

2. Has the statistical analysis been performed appropriately and rigorously?

Reviewer #1: Yes

3. Have the authors made all data underlying the findings in their manuscript fully available?

Reviewer #1: No

We will be happy to upload the minimal set of preprocessed data in a public repository upon acceptance of our manuscript and the full dataset can be obtained for non-profit purpose by contacting the corresponding author.

4. Is the manuscript presented in an intelligible fashion and written in standard English?

Reviewer #1: No

We have modified our manuscript substantially and hopefully it is now presented in an intelligible fashion.

5. Review Comments to the Author

Reviewer #1: Reviewer Comments for the Authors

We thank the reviewer for the comments and suggestions and we have tried to address all of them to the best of our abilities. The manuscript has been thoroughly modified as per the reviewer's suggestion and the revised manuscript will hopefully be acceptable to the reviewer. 

INTRODUCTION

Overall comments:

1. The authors could better summarize past literature on ERPs by highlighting and describing the work done in a few select seminal papers demonstrating N200/N2, P2, P300, and LPP’s respective importance and association with attention, food stimuli, and inhibitory control, which would provide more context for the chocolate stimuli they reference.

We thank the reviewer for the suggestion and have modified the ERP literature as per the reviewer's suggestion (page: 3 in the revised manuscript. )

2. The manuscript’s introduction could be more clearly laid out by having the authors group all their specific aims for the manuscript in a section at the end of the introduction instead of presenting them throughout the introduction. This would provide a better segue from their review of the literature, regarding what is known, to how they now approach filling in these research gaps with their specific aims for the current study.

We thank the reviewer for this comment and as per the reviewer's suggestion, we have moved the specific aims of the paper towards the end of the Introduction followed by elaboration on each of the objectives (pages: 4–5). 

3. Following the above comment, in some instances the aims of the paper seem to be repeated. It would be helpful if the authors could more succinctly summarize their aims and objectives perhaps grouping them as:

Whether behavioral desire ratings for craved foods translates to Willingness-To-Pay value measures

Prediction of Willingness-To-Pay value for craved food items

Efficacy of the Reversal Food Preference for the food craving induction for disliked foods

Neural correlates of food craving and its modulation (univariate and multivariate analyses)

We have modified the objectives as per the reviewer's suggestion and trimmed down the repetition of goals in the revised introduction. 

4. In addition to two previous points made above, the methods and techniques used to achieve each of these aims and objectives are sporadically referenced throughout the introduction as the aims are not consolidated in one specific section. By marrying each brief description of the approach used with each aim/objective the authors would be able to improve the clarity of their manuscript and better layout the experiments they detail in the Methods and Results sections of the manuscript.

We thank the reviewer for the suggestion and have now included one paragraph for each aim along with methods used to address the aim. The revised introduction has indeed become more concise and streamlined following the reviewer's suggestions.

5. The reversal of food preference is mentioned only at the end of the introduction. As it is highlighted in the title of the manuscript and a main topic of the manuscript, the authors should consider mentioning it sooner in their introduction and place more emphasis on its importance and relevance in their study, perhaps introducing it along with or after their summary on past literature on the regulation of craving.

We have now discussed about reversing of food preference towards the beginning of the introduction and also elaborated on it in the revised manuscript (page: 3)

Specific comments:

6. The wording, “progress from craving to spending,” on page 3 could be clarified, perhaps by using the word valuation instead. Here, the authors could also emphasize the progression and motivation behind acting on cravings, expanding on this idea as well.

We have reworded the sentence as per the reviewer's suggestion and expanded on objective valuation on craving (page: 3).

7. Similarly, the wording, “efficient public health policies” on page 3 could be improved by saying replacing “efficient” with “effective”.

We have changed the sentence as per reviewer's suggestion (page: 3).

MATERIALS & METHODS

Overall Comments:

8. The authors could describe the criteria for their sample better by providing both an inclusion and exclusion criteria for their participants.

We thank the reviewer for his comment, we have now included the inclusion criteria for the participant selection (page: 5).

9. The methodology and experimental timeline of the task are intermixed in the Methods section, making it hard to follow how the experiment was carried out. The authors could improve clarity on this by detailing their methods and measures used in the study prior to then separately outlining the timeline of the task in another section.

As per the reviewer's suggestion, we have described the task separately in the revised manuscript and specific details of behavioral and eeg experiment including timeline are included in the following subsections (pages: 5-7). Methods and measures used in the study have been moved to separate subsections followed by description of experiments (pages: 9–11).

10. The labels “Experiment 1” and “Experiment 2” that the authors used would benefit from having more descriptive names as Experiment 1 has no EEG component and is primarily behavioral, while Experiment 2 includes EEG with the task. The authors should consider renaming “Experiment 1” and “Experiment 2” to “Behavioral Experiment” and “EEG Experiment”, respectively.

 We have implemented the changes suggested by the reviewer (pages: 6–7)

11. It would be helpful if the authors specified what modification they made to the Becker-DeGroot-Marshak task when they discuss it on page 6 and include references to past literature validating this approach.

 We thank the Editor and Reviewer for pointing this issue out. To clarify, we did not modify the BDM process. However, Instead of considering BDM on a single trial from the last completed block (as used in the study by Glimcher et al. 2017), we randomly selected one out of 13 blocks and considered all trials in that block for the bonus calculation using standard BDM auction. We acknowledge that the way it was worded in the manuscript was misleading and we have now clarified this point into the manuscript with more detail (page: 7) and the revised manuscript now reads:

"Participants received Rs, 100 and a snack food item that he/she desires the most for their participation and a bonus amount, which was calculated using standard Becker-DeGroot-Marschak (BDM) mechanism [47]. Instead of considering BDM on a single trial from the last completed block [23], we randomly selected one out of 13 blocks and considered all trials in that block for the bonus calculation using standard BDM auction (see Supporting Information for details". 

12. Discussion of the goals of the study in the methods section should be omitted and moved into the introduction and discussion as appropriate.

The discussion goals have been moved to the introduction and discussion.

Specific Comments:

13. The authors should explicitly state what variables were used in their Spline Regression Models for clarity on page 9.

We have highlighted the explanatory variables used in our models (pages: 9–10). In the Spline Regression Model, we have used the desire values for determining the knots (page 10).

14. Sample sizes should be reported with (n=#) notation throughout the manuscript.

We have implemented the change suggested by the reviewer.

15. The authors should consider spelling out rather than abbreviating “viz.” on page 8.

We thank the reviewer for the comment. We now changed ‘viz.’ to ‘namely’.

16. The authors should include Table 1 in the Results section rather than the Materials and Methods section.

We have implemented the change as per reviewer's suggestion.

RESULTS

Overall comments

17. The authors should consider restructuring the results section by summarizing and comparing the results for all the different groups (DFE, LFE, NEC) for each of the analyses together instead of reporting the results by group in three different sections. This would cut down on repeated explanations of what the analysis is and what was done and relieve the need to have a summary at the end of the Results section as the main findings between the group comparisons are summarized in the respective analysis sections previously.

We thank the reviewer for the suggestion and have now restructured the Results section as per the suggestion making the results more concise (pages: 11–20).

18. The authors should also consider including change point detection plots for the non-exposed foods. This would help visualize the magnitude changes in bid and desire ratings for the food item used in the multisensory food exposure experience relative to the food items participants were not exposed to.

 We thank the reviewer for the suggestion. The change point detection plots for the non-exposed foods was previously included in the SI, but have now been included as Figure 5 in the main manuscript (page: 14)

Specific Comments:

19. For clarity the authors should more explicitly state which Chi-square test is for their behavioral experimental sample and which is for the EEG experimental sample on page 10.

We have used frequency Chi-square tests for independence for both behavioral and neural analyses (page: 9).

20. The authors should consider adding more descriptive title and axis labels to Figure 3 to distinguish the behavioral and EEG panels of the figure.

We thank the reviewer for pointing this out. Now we have included the titles of the subplots of Figure 3 to distinguish the behavioral and EEG experiments.

21. Additionally for each panel in Figure 3, the group comparison of FCQ-S scores would be easier to read and visually compare if the before and after bars for each group (DFE, LFE, NEC) were shown side-by-side.

 We agree with the reviewer and modified Figure 3, by plotting the before and after bars side-by-side for each of the groups (DFE, LFE, and NEC). 

22. Figure 4 would benefit from using different hues of the experimental group’s color to distinguish between the bid value and desire responses plotted for each group. Additionally it would be good to add visualizations of the statistical tests done to the plots to better highlight the change point findings reported in the results.

 We thank the reviewer for the comment. To visually distinguish between desire and bid value, we depicted these values with different hue colors for each of the groups. Also, p-values from the change point detection analysis corresponding to each of the changes are identified and depicted using the number of stars (‘*’). 

23. The authors should consider refraining from describing models with superlatives like “best model” on page 12, perhaps using words like “optimal” instead.

 We thank the reviewer for pointing this out. We have changed the ‘best model’ to the ‘optimal model’ in the manuscript (page 10 and page 15). 

24. The authors should consider adding axis labels and subplot titles in Figure 5 to clarify which groups what pre- or post-exposure model being shown in each panel. Additionally the authors should consider using a maker other than an asterisk in these plots, such as open circles, so the density of points can be better assessed as the asterisks are harder to distinguish from one another when they overlap.

We thank the reviewer for the comment. We have included the titles of each of the subplots and depicted each data point using an open circle instead of an asterisk for better visualization (see Figure 6 of the revised manuscript). 

25. Figure 7, 8, and 9 could be collapsed into one figure with the same legend, with separate panels for each of the different topological clusters EEG data was recorded from.

We thank the reviewer for the suggestion. We have incorporated all the suggested changes (please see Figure 8).

DISCUSSION

Overall comments:

26. The authors should be cautious when using the word craving throughout their discussion. Their results suggest that individuals do enter a state of craving, but from what they describe in their methodology and the results they present it is unclear if this food craving is specifically for the disliked food item or if it is that they are experiencing some form of a craving.

We thank the reviewer for pointing us towards this issue. We have now modified the discussion and replaced the word craving wherever applicable with "induce liking","desire", "liking for disliked food" and "Reversal".

Specific Comments:

27. The authors are unclear in part of their statement on page 21 that states, “... increase in desire does not always lead to WTP.” The authors should clarify whether they mean an increase or decrease of WTP.

We thank the reviewer for pointing out the mistake in the above sentence. We have corrected it and the modified sentence now reads "... increase in desire does not always lead to an increase in WTP.”

---

## [Editor Report · Decision Letter 1]

4 Jul 2023

Reversing Food Preference Through Multisensory Exposure

PONE-D-22-32995R1

Dear Dr. Das,

We’re pleased to inform you that your manuscript has been judged scientifically suitable for publication and will be formally accepted for publication once it meets all outstanding technical requirements.

Kind regards,

Joydeep Bhattacharya

Academic Editor

PLOS ONE
---

## [Editor Report · Acceptance letter]

11 Jul 2023

PONE-D-22-32995R1 

Reversing Food Preference Through Multisensory Exposure 

Dear Dr. Das:

I'm pleased to inform you that your manuscript has been deemed suitable for publication in PLOS ONE. Congratulations! Your manuscript is now with our production department. 

Kind regards, 

on behalf of

Dr. Joydeep Bhattacharya 

Academic Editor

PLOS ONE